# A Municipality-Based Approach Using Commuting Census Data to Characterize the Vulnerability to Influenza-Like Epidemic: The COVID-19 Application in Italy

**DOI:** 10.3390/microorganisms8060911

**Published:** 2020-06-16

**Authors:** Lara Savini, Luca Candeloro, Paolo Calistri, Annamaria Conte

**Affiliations:** National Reference Centre for Veterinary Epidemiology, Programming, Information and Risk Analysis, Istituto Zooprofilattico Sperimentale dell’Abruzzo e Molise “G. Caporale”, 64100 Teramo, Italy; l.candeloro@izs.it (L.C.); p.calistri@izs.it (P.C.); a.conte@izs.it (A.C.)

**Keywords:** COVID-19, commuting census data, municipality-specific infection contact rate, vulnerability, infectious disease modeling

## Abstract

In February 2020, Italy became the epicenter for COVID-19 in Europe, and at the beginning of March, the Italian Government put in place emergency measures to restrict population movement. Aim of our analysis is to provide a better understanding of the epidemiological context of COVID-19 in Italy, using commuting data at a high spatial resolution, characterizing the territory in terms of vulnerability. We used a Susceptible–Infectious stochastic model and we estimated a municipality-specific infection contact rate (β) to capture the susceptibility to the disease. We identified in Lombardy, Veneto and Emilia Romagna regions (52% of all Italian cases) significant clusters of high β, due to the simultaneous presence of connections between municipalities and high population density. Local simulated spreading in regions, with different levels of infection observed, showed different disease geographical patterns due to different β values and commuting systems. In addition, we produced a vulnerability map (in the Abruzzi region as an example) by simulating the epidemic considering each municipality as a seed. The result shows the highest vulnerability values in areas with commercial hubs, close to the highest populated cities and the most industrial area. Our results highlight how human mobility can affect the epidemic, identifying particular situations in which the health authorities can promptly intervene to control the disease spread.

## 1. Introduction

On December 31 2019, the Chinese Country Office of the World Health Organization (WHO) was informed about cases of pneumonia of unknown etiology in residents of Wuhan City, Hubei Province of China [1]. Later defined as a new disease (COVID-19) caused by a novel coronavirus (SARS-CoV-2), the epidemic was declared by WHO a public health emergency of international concern on January 30 and a “pandemic” on March 11.

In February 2020, after the first outbreak of infection detected in Codogno municipality, Lodi province, of Lombardy region, Italy became the epicenter for COVID-19 in Europe and on February 22 the Italian Government imposed a lockdown in hotspot areas in Lombardy and Veneto regions. On March 8, to further contain the spread of the virus, the red zone was extended to the whole area of Lombardy and to 14 other provinces of northern Italy [2,3,4]. On March 9, in response to the growing epidemic, emergency measures that restricted population movement (except for essential work categories and health reasons) were extended to the whole country. As of June 2, 233,515 positive cases were recorded, and 33,530 people died, making Italy the sixth country in the world by the number of total cases, after the United States of America, Brazil, the Russia, Spain and the United Kingdom and the third in the world by the number of deaths.

Human mobility represents a crucial element to be considered in modeling human infectious diseases and the main factors influencing mobility patterns and its magnitude depend on the scale considered (global, national, local). At the global level, the study of air traffic connections may provide good indications for predicting a worldwide spread [5,6]. On a local scale, other types of movement must be considered. Open-data resources and data-driven models offer many opportunities to improve governments’ responses to the new epidemic and various studies have recently been conducted to assess how different human mobility data, such as Google’s mobility data or data collected via mobile phone, can guide government and public health authorities to evaluate the effectiveness of measures to control the COVID-19 spread [7,8,9]. However, commuting, defined as the daily movements from residence to work or school, is certainly the most relevant and widely studied factor to describe spatial mobility in local models [10,11,12].

In this paper, we analyze the commuting flows in Italy, using census data (ISTAT 2011) [13], in order to assess its contribution in spreading the COVID-19 at the beginning of the epidemic.

The aim of our analysis is to provide a better understanding of the epidemiological context of COVID-19 in Italy, and to characterize the territory in terms of vulnerability either at the local or national level. The objective is not to accurately estimate the number of cases or the magnitude of the epidemic in absolute terms, but to understand how the disease can spatially and temporally spread countrywide.

## 2. Materials and Methods

The study is organized to first evaluate the use of commuting data as a risk factor of COVID-19 spreading in Italy, calculating Social Network Analysis (SNA) centrality measures at the province level and performing a data correlation analysis. Then we examined the underlying mechanisms of propagation using a stochastic Susceptible–Infectious (SI) model mainly driven by the commuting network and by a revised infection contact rate parameter. We modeled a municipality-specific infection contact rate to capture the permeability to the disease of each municipality, considering the population at different times of the day and considering the characteristics of municipalities as attractors of commuters or displacing their workforce elsewhere.

### 2.1. Demography and Commuting Network

To analyze the Italian commuting network, we used census data collected in 2011 [13]. All data is obtained at the municipality level. About half of the 60 million people living in Italy declared a daily movement to their usual place of study or work.

After adjusting the geographical dataset of the Italian municipalities according to the modifications that occurred after 2011, the matched commuting dataset contains 7,915 municipalities, a resident population of 60,340,328 and 28,805,440 commuters, within (17,497,742) and between (11,307,698) municipalities. Commuting flows directed to or coming from abroad are not considered in the analysis. Figure 1a,b shows the distributions of the Italian population and commuters, respectively.

A commuting network is generated by creating a direct weighted edge between two nodes, represented by the municipalities of origin and destination. The weight indicates the number of commuters traveling on that connection in a typical working day (Figure 2a). The resulting network is composed of 7915 nodes and 539,223 edges. Then, the commuting network is rescaled at the province level (the lowest NUTS level (Nomenclature of Territorial Units for Statistics), NUTS 3) in order to compare the results of the network analysis with the COVID-19 cases as recorded at the province level by the Dipartimento della Protezione Civile [14] and archived on GitHub [7]. The rescaled network has a size of 107 nodes and 3310 edges (Figure 2b).

The rescaled commuting network is analyzed by calculating the centrality measures commonly used in epidemic modeling [15,16,17,18]: degree (in-out), strength (in-out), betweenness, both for the global network and for subnetworks (incoming and outgoing commuters greater than 50,100 and 1000).

### 2.2. SI Model to Evaluate the COVID-19 Spread Dynamics

#### 2.2.1. Theoretical Geodemography Framework

The model is based on the commuting network at the municipality level. In a municipality, we have the resident population divided into noncommuting (R), commuting within the municipality (C_r_) and commuting outgoing the municipality (C_o_). Commuting affects the number of people present in a municipality during the different times of the day, significantly modifying the registered resident population. If we define C_i_ as the nonresident population commuting into a municipality, we have that the real population in a specific time of the day is given by R +C_r_ – C_o_ + C_i_ (Figure 2a). On the right side of Figure 3, three municipalities (A, B, C) with their registered residents are shown; the left side shows the actual population present, following in and out commuting during the day. In Municipality C, the resident population equal to 10, becomes 4 during the day, due to a prevalent component of C_o_. Conversely, the population in B significantly increases during the day compared to the resident population, due to the higher C_i_ component (from 10 to 16). Municipality A has a balanced population during the entire day, having an equivalent C_o_ and C_i_ components.

The typical working day is divided into two parts based on people’s daily contact dynamics: “high activity” time during which contacts are facilitated by the social common activities (e.g., work, school, sports and similar) and “low activity” time in which the activities are reduced or stopped (e.g., during night or sleeping time).

The “high activity” and “low activity” times of the day, in combination with the resident and commuters’ populations, determine different levels of contact (and therefore of infection) between municipalities and between individuals within the municipality.

#### 2.2.2. Model Implementation

A Susceptible–Infectious (SI) compartmental model is implemented to simulate disease spread due to the commuting between municipalities, taking into account not only the absolute commuters values, but including the influence of the typical structure of a day (high and low activity) as described in the previous paragraph. It is an agent-based model where a subject susceptible can become infectious if living or working within an infected population. The following ODEs system describes the model:
(1)$$\left\{\begin{array}{c} S_{i}^{m}\left(t+1\right)=S_{i}^{m}\left(t\right)-\beta_{i}^{m}\frac{I_{i}^{m}\left(t\right)S_{j}^{m}}{N_{i}^{m}}\;\\ I_{i}^{m}\left(t+1\right)=I_{i}^{m}\left(t\right)+\beta_{i}^{m}\frac{I_{i}^{m}\left(t\right)S_{j}^{m}}{N_{i}^{m}}\;\;\;\; \end{array}\right.$$
where S is the susceptible population, I is the infected population, N is the population size, β is the infection contact rate, *i* indicates the municipality, *m* indicates the time of the day (“high active” or “low active” time) and *t* is a specific day; the equation describing the transition state of each individual, from susceptible (s) to infected (i), follows a Bernoulli distribution:
(2)$$P\left(s\;\left(t\right)\to i\left(t+1\right)\right)=\;\mathrm{Bern}\left(\beta_{i}^{m}\frac{I_{i}^{m}\left(t\right)}{N_{i}^{m}}\right)$$

Therefore, we assume a homogeneous mixing of the population adopting two infection contact rates for each municipality to take into account the time of the day (“high active” or “low active” time) and the related variation in population density due to commuting.

##### Infection Contact Rate Parameter Modeling

Assuming the same infection contact rate β for the whole country implies the disease spreads with the same strength everywhere, even inside those areas highly different spatially, demographically and in terms of commuting systems.

Our approach defines a different β for each municipality in different times of the day as it takes into account the variation in population density due to commuting: β should be higher in densely populated places and lower in poorly populated ones. Moreover, in our theoretical framework, we assume two overlapping populations staying in the same place during the day (but in two distinct times) as shown in Figure 4. While the subpopulation is given by commuting within the municipality (C_r_) and noncommuters (R) lives within the place (L) during the whole day, incoming (C_i_) and outgoing (C_o_) commuters contribute to two different populations during the “high activity” and “low activity” times. In this scenario, during the “high activity” time the total population staying in the area is C_r_ + R + C_i_, while during the “low activity” time, it is C_r_ + R + C_o_.

The ratio between the population staying in a municipality in each time of the day and its surface (area in km^2^) giving the population density per time is:
(3)$$D^{m}=\frac{P^{m}}{AreaL}$$
where *m* is the “high activity” (h) and “low activity” (l) times, and P^m^ = (R +C_r_) + C_i_ if *m* = h, or P^m^ = (R +C_r_) + C_o_ if *m* = l.

In Figure 3, assuming that the area of Municipality C is 100 km^2^, its corresponding population density during the “low activity” time is 10/100, while its population density during the “high activity” time is 4/100. We adjust D^m^ to account for commuters contributing to disease spread, using a (multiplicative) commuting factor (Cf), according to the following equation:
(4)$$D^{m*}=D^{m}Cf\;\mathrm{where}\;Cf=\frac{2C^{m}}{C^{m}+R}$$

Cf increases the population density if the C^m^ component (i.e., the commuters given by C_i/o_ + C_r_) is greater than the noncommuters population (R) and reduces it if P^m^ is mainly made of R. 

In this way, we assume that commuters have a greater weight than residents have in driving the disease spread within the municipalities. In Figure 3, Municipality C, during the “high activity” time, has a population density D^h^ = 4/100, having C^h^ = C_i_ + C_r_ = 2 and R = 2, the corresponding Cf = 2*2/(2+2) = 1 and thus the adjusted population density D* = D^h^ * Cf = D^h^ = 4/100 is the same of the nonadjusted population density. Differently, during the “low activity” time, the C population density is D^l^ = 10/100, being C^l^ = C_o_ + C_r_ = 8 and R = 2. The corresponding Cf = 2*8/(8+2) = 8/5 and the D* = D^l^*8/5 = 16/100.

Hence, Cf made the population density higher during the low activity time (turning from 10/100 to 16/100) because of its high number of commuters (C_o_) respect to noncommuters’ population (R).

The following Figure 5 shows the effect of the adjustment applied to each municipality population density for low and high activity times (a) and the adjusted population density distributions using the logarithmic scale (b).

Since the distribution of the adjusted population density is lognormal (Figure 5b) we use the following scaling function to get a Z measure for low and high activity time:
(5)$${\mathrm{Zscore}}^{m}=\frac{\log\left(D^{m*}\right)-\mathrm{mean}\left(\log\left(D^{m*}\right)\right)\;}{\mathrm{sd}\left(\log\left(D^{m*}\right)\right)}$$

As expected, Zscore^m^ are highly correlated, meaning locations with high values of Zscore for “high activity” time also have high values of Zscore for “low activity” time. However, locations diverging from this general tendency have a peculiar meaning.

Figure 6a shows the distribution of paired Zscores for each municipality, using two colors to code the difference between “high activity” and “low activity” times Zscores. Points (municipalities) with positive values of the difference (red dots), regardless of how dense the population is, are places having a higher incoming commuting component in comparison to the outgoing (Municipality B in Figure 3 is an example of this kind of structure). These kinds of municipalities are a sort of “attractors” of the workforce or students, like municipalities with developed industrial poles or universities. Figure 6b shows the Abruzzi region map (in central Italy) of such a difference. It is evident the presence of the industrial pole of Atessa municipality in the southern region (dark red municipality). Blue dots in Figure 6a represent instead places having negative values of the difference. In such places, commuters working outside their own municipality are the majority of the present population (as for Municipality C in Figure 3). These kinds of municipalities are typically emptied during the “high activity” time, because, lacking developed manufacturing, production, or tertiary sectors, they make their workforce available for attractors. Figure 6c shows the geographic distribution of the difference for the Lazio region (neighboring Abruzzi region in central Italy). It is evident the blue ring around Rome, made of municipalities where the majority of workers are outgoing commuters (to Rome). Municipalities having difference values close to zero (white dots) have both high or both low values of Zscores (like Municipality A in Figure 3).

The effective infection contact rate (β_0_ = 0.244) is estimated from the temporal evolution of the cases observed between February 26 and March 6, assuming an exponential infection growth [19]. We use the following equation to scale the daily value of the effective infection contact rate:
(6)$$\beta^{m}={\left(1+\beta_{0}\right)}^{\frac{H^{m}}{H}}-1$$
where H is the number of hours in which the hourly β is greater than zero (we assume the hourly β = 0 during sleeping time, 10 h/d) and H^m^ is the number of hours relative to “high” and “low” activity times (H = 14 and H^m^ = 11: during “high activity” time, 3; during “low activity” time). Thus, being β_0_ = 0.244, β^m^ = {0.1871; 0.0479}.

Finally, we define the infection contact rate for each municipality as depending on the population staying at different times of the day:
(7)$$\beta_{i}^{m}={\;\mathrm{Zscore}}_{m}\left(\frac{\mathrm{vc}*\log\left(D_{i}^{m*}\right)}{2{*\beta }^{m}}\right)+\beta^{m}$$
where $D_{i}^{m*}$ is the adjusted population density according to commuting flows in municipality *i* and time *m* and *vc* is the coefficient of variation of the $\log\left(D_{i}^{m*}\right)$ distribution.

The daily β per municipality is expressed as the product of the $\beta^{m}$:
(8)$$\beta_{i}=\left(\beta_{i}^{\mathrm{high}}+1\right)\left(\beta_{i}^{\mathrm{low}}+1\right)-1$$

##### Model Simulation Scenarios

In order to highlight the different perspectives of the developed model and the potential of its application, three scenarios are considered:

**Scenario 1.** COVID-19 spreading at the municipality level for the entire Italian territory between February 26 and March 6. We focused on the first 10 days of the epidemic because most of the impact that commuting may have had is before the application of the lockdown measures. The model assumes that no restrictions on human mobility are put in place at the beginning of the epidemic. Inside each province with confirmed COVID-19 cases at the starting time (February 26), the infected people are assumed to be randomly distributed. Results are then compared at the province level with the official data on confirmed cases.

**Scenario 2.** Local spreading (during the first 21 days of the epidemic). This scenario is implemented to explore and compare the spread patterns in different regions (rather than validate the predictive capacity), to better understand how human mobility as a spread driver can affect the epidemic. Lombardy, Abruzzi and Basilicata regions are chosen based on the level of infection (high, medium and low) observed during the epidemic. In this scenario we assume that the infection starts at the municipality level (the first infected municipality notified in the region is considered as seed); no restrictions on human mobility are put in place at the beginning of the epidemic; the regional network is closed to external commuting exchanges. A longer timeframe is chosen to better explore and compare the different spread patterns.

**Scenario 3.** Local spread in the Abruzzi region is simulated (during the first 14 days of the epidemic) considering each municipality as a seed for each simulation. This scenario aims at defining the vulnerability of each municipality to a new epidemic (or a new epidemic wave) and it gives an indication about the potential of each municipality to be the index point of new infections.

The choice of the Abruzzi region has only an illustrative purpose. We assume that the infection starts in turn in each municipality so as to assess the weak points of the whole region.

All simulations use the infection contact rate β described in the previous paragraph, assuming that the number of plausible active cases (K) is 10 times the number of those officially detected, as reported by the Italian Institute for International Political Studies [20]. In Table 1, all the simulation parameters used in the three scenarios are listed.

During the first five days of the week, as working days, β is calculated considering the “high activity” time for the commuters adjusted population (R + C_r_ − C_o_ + C_i_), whereas the “low activity” time is applied to the resident population (R + C_r_ + C_o_). The opposite occurs during the weekend (the last two days), to take into account that the resident population becomes more active than that determined by commuter flows.

The model is run at the municipality level. For each node (municipality), for each individual of the node, and for each temporal step t of a day, the model recalculates the status (in terms of S and I) of the source and destination nodes for the next temporal step by following the equations above described.

The pseudo model algorithm is coded as follows:
**Algorithm 1** The pseudo model algorithm*#constants shared between scenarios**K = 10*$\beta_{i}^{m}$*#constants within scenario**SIM (number of simulation)**Deep (number of days to be simulated)**NTW (Commuting network: weighted links among municipalities of the corresponding scenario)**I (number of officially infected)**For each simulation**Sample infected according to scenario**For each day**If during weekend, switch the correspondence between active period and population**For each sub-day period*  *For each infected municipality**      find new infected according to ODE equations**End*

The model is stochastically implemented in R-software (Version 3.6, R-Foundation for Statistical Computing, Vienna, Austria); “doparallel” R package is used for parallelizing the simulations [21]. Figures are created in R using “sp” and “ggplot2” libraries [22,23], in ArcMap 10.5 ESRI and Cytoscape (Version 3.2.1) programs.

## 3. Results

### 3.1. Analysis of the Commuting Network

A Pearson’s correlation matrix was calculated among the SNA metrics of the commuting network (and subnetworks) and COVID-19 cases (at different times of the epidemic), (Figure 7a). 

Degree measures and COVID-19 cases showed a significant correlation (*p* < 0.05). In particular, the degree calculated for the subnetwork built on the basis of incoming and outgoing commuters greater than 50 (Deg50) and COVID-19 cases (as of March 26) have the highest correlation value equal to 0.72 (black-boarded square in Figure 7a).

Figure 7b represents a scatter plot between Deg50 and COVID-19 cases on the logarithmic scale. To capture the commuting behavior of each province (node) the scale color from green to red is used to characterize the in-strength (incoming commuters from lower to higher) and the symbol size characterizes the node in terms of out-strength (outgoing commuters from smaller to bigger).

It is evident that the northern provinces, most affected by the disease, are also those characterized by a high degree (flow of commuters among multiple provinces) and high strength (incoming and outgoing commuters exchanged) while the provinces of central and southern Italy, with a lower number of cases, are mainly characterized by lower degree and in-strength values.

### 3.2. Infection Contact Rate Parameter Estimation 

The infection contact rate β values are calculated based on commuting data and population size in each municipality. Figure 8a shows the β values per municipality (quantile aggregation), while Figure 8b shows the statistically significant hot spots, cold spots and spatial outliers of β values using the Anselin Local Moran’s I statistic [24].

Although the map of infection contact rates mainly reflects the resident population density, the municipality-specific infection contact rate captures the disease permeability of each municipality, considering the population in different moments of the day and depicting the characteristics of municipalities as attractors of commuters or as displacer of its workforce elsewhere.

The clusterization of areas with similar characteristics in terms of vulnerability to the introduction and spread of the disease is highlighted by Moran’s analysis (Figure 8b). There are large areas characterized by the homogeneous presence of high β values (pink areas) where the introduction of the disease would inevitably lead to a spread more difficult to control due to the simultaneous presence of connections between municipalities and the high population density. On the contrary, the large areas with low β levels (light blue) represent areas in which the disease spreads slowly (e.g., Alps and Apennine areas, Basilicata, Sardinia, the southern part of Tuscany and Molise). The red areas constitute potential outliers which, despite a high β value, would hardly expand rapidly the disease in the surroundings which have instead low β values. In this way Figure 8b, grouping similar municipalities’ values gives an immediate and overall picture of the Italian territory in terms of higher or lower susceptibility to an epidemic.

### 3.3. SI Model to Evaluate the COVID-19 Spread Dynamics

Three scenarios are evaluated as listed in Table 1: COVID-19 spreading at the municipality level for the entire Italian territory between February 26 and March 6 (Scenario 1); local spreading (during the first 21 days of the epidemic) in Lombardy, Abruzzi and Basilicata regions (Scenario 2); local spread in the Abruzzi region (during the first 14 days of the epidemic) considering each municipality as a possible seed; origin of the infection (Scenario 3).

#### 3.3.1. Scenario 1

The number of infected provinces during the studied period increased from 29 to 92 and only 15 were still free from COVID-19 as of March 6. The number of infected people increased from 625 to 5,699. Considering the results at the national level, the model was able to correctly catch the observed cases’ trend (Figure 9a), while the observed increase of infected provinces was more rapid than the estimated one (Figure 9b).

Pearson’s correlation coefficient between observed and estimated infected people (median value) at the province level, at the end of the period, was equal to 0.92.

Figure 10 shows the agreement between observed and estimated infected provinces as of March 6. Provinces’ color represents the estimated probability of being infected, red points indicate the observed infected provinces at the end of the period, green points represent the observed noninfected provinces at the end of the period, dashed provinces are the seeds at the beginning of the period.

A threshold of p = 5% on the percentage of simulations with at least one infected individual per province was chosen to evaluate the model’s performance in predicting the infection status of each province at the end of the period. The comparison with observed official status as of 6 March resulted in a number of True Positive (TP) = 85, True Negative = 6, False Positive (FP) = 9 and False Negative = 7. This leads to a Sensitivity = 92.4% and a Specificity = 40%, for a total accuracy of 85%. Figure 11a,b shows the comparison between observed and estimated infected people (upper 95% confidence interval when a province is turned into infected in the simulations), respectively, excluding the seeds.

As far as the seeding sites concern, Figure 12 shows the comparison between observed and estimated infected people in each seed province. For most of the cases the observed values fall into the 95% confidence interval of the estimates. Results are grouped in four panels depending on the magnitude of the observed COVID-19 epidemic.

#### 3.3.2. Scenario 2

The differences observed among the spread patterns in high (Lombardy), medium (Abruzzi) and low (Basilicata) affected areas might be explained by different infection contact rates and different commuting systems.

Figure 13a–c compares β values, degree measure and the estimated probability of being infected, respectively, for each municipality. Starting from one seed in each region (the first municipality notified as infected), for a time window of 21 days, the disease seems to follow different patterns: in Lombardy, where β values are more homogeneous, the disease extends in wideness; in Abruzzi region the disease spreads along the Adriatic coast, driven by the β parameter higher in this corridor; in Basilicata, the disease remains confined to the point of origin because the region has neither high β values nor high connections.

Moreover, the differences between the three regions have been evaluated through the number of estimated infected people and infected municipalities (Figure 14a,b). The pattern of infected people is similar for Abruzzi and Lombardy (red and green line, respectively), but different from Basilicata (green line). However, the differences are more evident if we consider the number of infected municipalities that grows more rapidly in the case of Lombardy (Figure 14b). The speed with which the municipalities in Lombardy become infected is higher than in Basilicata and Abruzzi due to the connections underlying the commuting network (Figure 14c). It is noteworthy that Lombardy has 28% of municipalities with a degree greater than 140, while in Abruzzi and Basilicata regions 95% and 99% of municipalities, respectively are below a degree of 60.

#### 3.3.3. Scenario 3

In this scenario, each municipality is considered in turn as seed. The vulnerability of the Abruzzi region is calculated as the number of infected individuals (95th percentile) and infected municipalities that each municipality (seed) causes in the region (excluding itself) (Figure 15a,b). Figure 15c shows the ratio between the number of cases caused outside the municipality and the number of cases caused inside the municipality (× 100) that may be interpreted as the tendency of each municipality to act as a destination or origin of infection for the other territories.

## 4. Discussion and Conclusions

Human mobility data have been largely used for modeling the spread of infectious diseases both at global [2,6,10,11,25,26] and national levels [27,28,29].

Data on commuting, defined as the daily local movements from home to work location or schools, have been used in the study of the epidemiology of infectious disease to a lesser extent. Recent use of these data for modeling the COVID-19 epidemic in Italy has been published by Gatto et al. [12] and Vollmer et al. [8].

In Italy, the currently available data on commuting are census data, with the advantage of being structured, open-source and representative of the entire Italian population. It is updated every 10 years does not seem to affect the spatial patterns of human mobility [7,12], thanks also to the stability of the production systems and persistence of attractive poles (schools, offices, etc.) in the same places.

The use of data at a higher spatial resolution (municipality level) allows highlighting peculiar situations in which public health authorities may promptly intervene to control the spread of disease. For this reason, we have introduced within a metapopulation model, driven by the commuting network, a municipality-based infection contact rate able to capture the variability between municipalities in terms of population density and commuting system.

To have a model suitable in the first phase or in the case of the resurgence of the epidemic (when the population is fully susceptible) we used a simple Susceptible–Infectious model in which more importance was given to model the infection contact rate, neglecting the Exposed, Recovered and Dead compartments. Indeed, we only consider two epidemiological parameters, the number of (undetected) cases per each officially detected case (K) and the infection contact rate β. We made the assumption of K being equal to 10 [20]. Although more accurate estimation could soon be available, our results in terms of correlation and patterns are not affected by this parameter because it is a proportionality factor. Although we have reduced the number of parameters of the epidemiological model from one side, we have introduced other variables to model the socio-demographic and commuting aspects.

The revised calculation of the infection contact rate β, based mainly on the resident population density, also incorporates the commuting component in each municipality, highlighting its characteristic of being an attractor of commuters or a displacer of workforces toward elsewhere. When areas with high β values are contiguous and significantly clustered, disease permeability tends to be greater. Lombardy, Veneto and Emilia Romagna (situated in northern Italy), which together made up 52% of all Italian cases (as of May 7, 2020), are effectively clustered with similar and high β values (Figure 8b).

The revised municipality-based β can be generalized to any other epidemic that responds to the assumptions made for its calculation.

The simulation model was applied considering three different scenarios. The first scenario (Scenario 1), considering COVID-19 spreading at the municipality level for the entire Italian territory between February 26 and March 6, was used to assess the model accuracy.

At the national level, the model estimates the trend of infected people similar to the observed trend of cases (Figure 9a), despite the introduction of the β variability. As far as the number of provinces involved concerns, the estimates are more variable. In particular, the model estimates a growth rate of the infected provinces lower than the observed one (Figure 9b). This might be due to the uncertainty about the real number of provinces already infected at the beginning of the period. If a higher number of initially infected provinces had been considered, the outcomes of the model would have been more similar to the observed ones.

Considering the capacity of the model to correctly classify a province as infected at the end of the observation period, despite a global accuracy of 85%, the model failed in classifying 16 provinces. However, the misclassification may be due not only to the model capacity, but also to the influence of uncontrolled factors such as errors in the observed data, timing in the notification of cases, ability to identify the disease, containment local measures, long-distance displacement of people from infected areas to noninfected areas.

As for the false negative (FN) provinces, the model failed in identifying as infected, provinces in which the number of observed cases at the end of the study period was actually very low (from one to three); only one province out of seven never turned out infected in any simulation. Furthermore, most of the FN provinces are located in southern Italy, an area that was affected by the massive return of university students from the North, after all schools were closed. The nine false positive (FP) provinces were officially detected as positive a few days after the considered period.

The outcomes of the model were compared with the observed number of COVID-19 cases for the initially infected provinces, grouped by different virus circulation levels (from low to high). For some provinces, such as Palermo (PA), Savona (SV), Padua (PD), Milan (MI) and Lodi (LO) the model overestimates the cases, whereas in few others, like Pesaro–Urbino (PU) and Parma (PR) the number of cases was underestimated (Figure 12). One of the possible explanations for this disagreement may be found in the application of control measures by local authorities, which anticipated the measures of the central government. On the other hand, the model was able to estimate quite precisely the number of cases in several provinces (either at low or high virus circulation level). Among these, Bergamo (BG) province was one of the Italian provinces that suffered more for the COVID-19, with more than 10,000 cases and almost 3,000 deaths. This would lead one to think that, in the province, the control measures were not strictly applied at the very beginning of the epidemic. It is important to note that the limitation of the model in correctly estimating the magnitude or the extent of the epidemic also depends on the difficulty of including in the model the establishment of the community (hospitals, health care settings, working) or household clusters of infection.

The second scenario (Scenario 2) was developed to explore and compare the spread pattern in three different regions. The chosen regions (Lombardy, Abruzzi and Basilicata) are characterized by different epidemiological conditions. When the outcomes for the three regions are compared, the population density and level of industrialization must be taken into account. The mobility of workforces in Lombardy, which is one of the major economic driving areas in Europe, is much greater than in the other two regions. The differences among these regions are more evident when the number of infected municipalities is considered (Figure 14b), being the infection spreading across municipalities directly linked to the connections underlying the commuting network. Indeed, in the case of Lombardy, about 50% of the municipalities are connected to more than 100 municipalities (against 3% in Abruzzi and Basilicata regions; Figure 14c).

In the last scenario (Scenario 3) the local spread in the Abruzzi region is estimated (during the first 14 days of the epidemic) considering each municipality as a seed for simulation. This scenario has the purpose of identifying those municipalities more vulnerable to the virus introduction and those playing a major role in spreading the infection. The vulnerability of the Abruzzi region is calculated as the number of individuals that each municipality (seed) causes in the region and the number of infected municipalities (Figure 15a,b). In addition, the ratio between the number of cases caused outside the municipality and the number of cases caused inside the municipality may provide a useful hint about the risk category (capability to infect rather than become infected), of each municipality (Figure 15c). The obtained maps, at the municipality level, provide the decision-makers with useful information on where mobility restriction measures should be focused to have the strongest effects on transmission reduction. The highest vulnerability values can be observed in areas with commercial hubs, close to the highest populated city of the region, Pescara (Figure 15b, the darker blue area on midcoastal line) and the most industrial area of the region, in the Sangro Valley (Figure 15b, in the south of the Abruzzi region), where many medium and big factories are present.

Our approach, therefore, provides decision-makers with geographically detailed metrics to evaluate those areas at major risk for infection spreading and for which restrictions on human mobility would give the greatest benefits. It can provide risk maps in which health administration can modulate the application of strong lockdown measures, evaluating in advance the effects on reducing the spread of the infection.

This approach is particularly useful not only at the beginning of the epidemic but also in the last phase, when the risks deriving from the gradual lockdown exit strategies must be carefully evaluated. In fact, the major risk in this latter phase is the resurgence of infection transmission through a progressive reopening of the productive systems.

The analysis of daily human mobility patterns for working reasons is clearly providing a well-detailed picture of the areas and productive systems more at risk of sustaining a restart of the epidemic. This study although, based on the current epidemic, will provide useful elements for other influenza-like epidemics that might happen in the future, helping health authorities to implement and direct the right interventions measures.

## Figures and Tables

**Figure 1 microorganisms-08-00911-f001:**
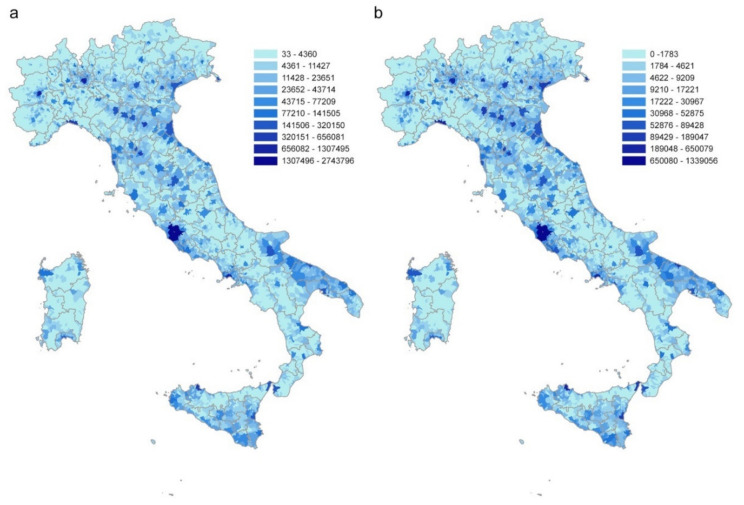
Distribution of (**a**) the Italian population and (**b**) daily commuters at the municipality level. Source ISTAT.

**Figure 2 microorganisms-08-00911-f002:**
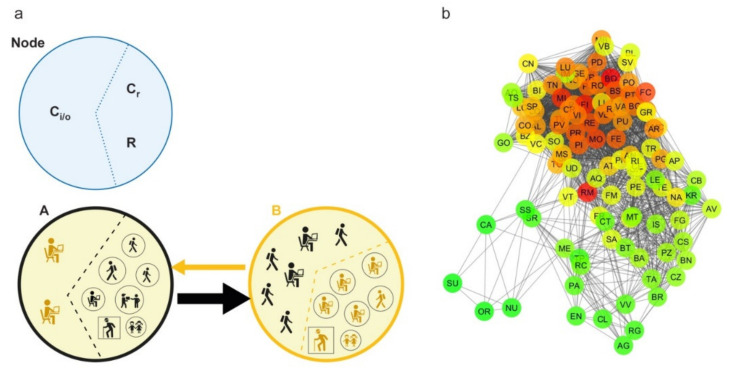
Commuting network structure. (**a**) All categories of commuters and noncommuters that characterize each node of the network on a typical working day are represented: commuters inside the node (C_r_), incoming and outgoing commuters (C_i_ and C_o_, respectively) and residents who are not commuters (R). The edge between two nodes, from a source to a destination, is represented by an arrow (direction) and its size is proportional to the number of commuters moving daily. (**b**) Graph representation of the commuting network at the province level (undirected for display purposes). The scale color from green to red is based on the node degree value.

**Figure 3 microorganisms-08-00911-f003:**
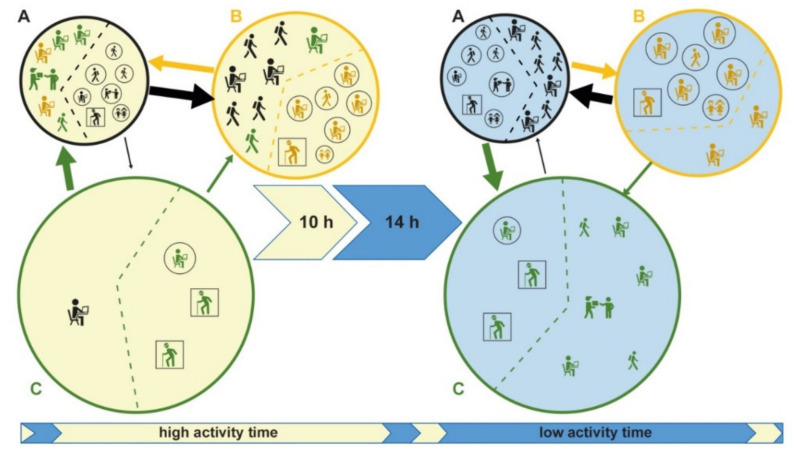
A typical working day description considering three different municipalities and populations. Population changes due to commuting in two different times of the day (high activity and low activity). The municipalities, A, B, and C are displayed by circles in black, yellow, and green; the icons, colored according to the circles, represent the resident populations in the 3 municipalities; squared icons represent noncommuters resident (R) and circled icons commuters inside the node (C_r_); the arrows represent commuters moving daily from a source to a destination (C_o_ and C_i_) and the arrow’s size is proportional to the number of moving commuters. Municipality C, with a population of 10, decreases to 4 during the day, due to a prevalent component of outgoing commuters (C_o_ = 7). Conversely, the population in B increases significantly during the day, due to the higher incoming commuters (C_i_ = 8). Municipality A has a balanced population during the entire day, having an equivalent C_o_ and C_i_ components.

**Figure 4 microorganisms-08-00911-f004:**
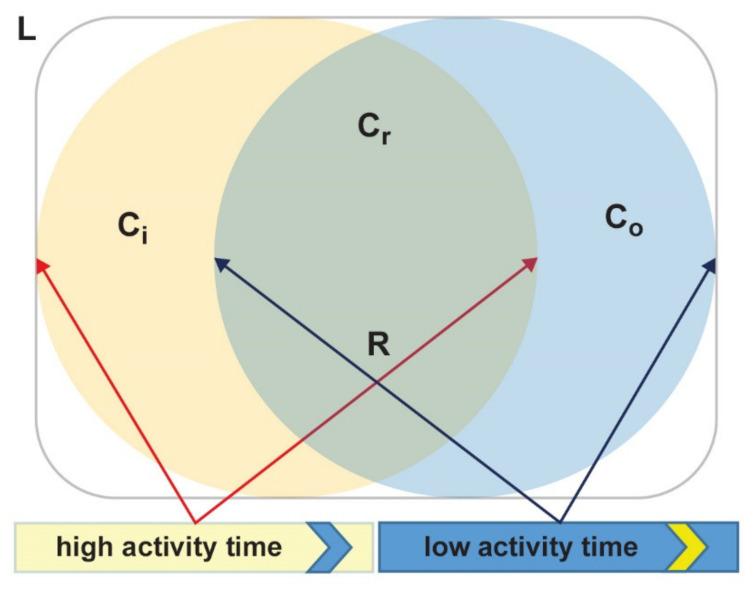
Within-day evolution of people staying in the same place L. Individuals staying inside the overlapping area during the whole day are commuting within the municipality (C_r_) and noncommuters (R): C_r_ + R. During the high activity time (yellow circle), incoming commuters (C_i_) stay with R and C_r_, while during low activity time (blue circle), outgoing commuters (C_o_) come back and incoming commuters (C_i_) go away and individuals inside the municipality are: C_r_ + R + C_o_.

**Figure 5 microorganisms-08-00911-f005:**
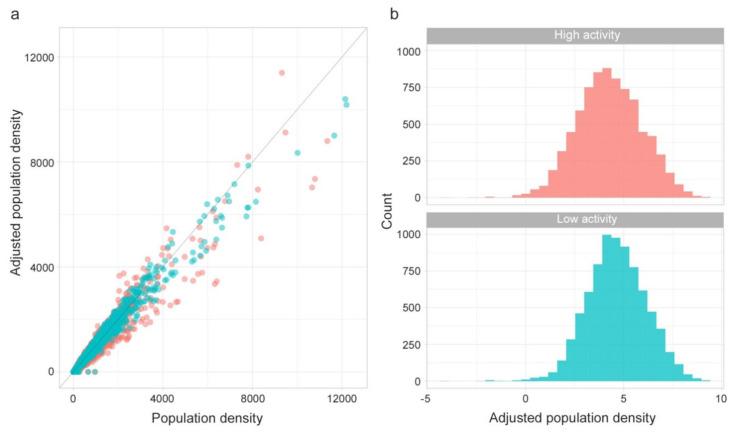
(**a**) Population density adjustment and (**b**) adjusted population density distribution for low activity time (blue) and high activity (red) time.

**Figure 6 microorganisms-08-00911-f006:**
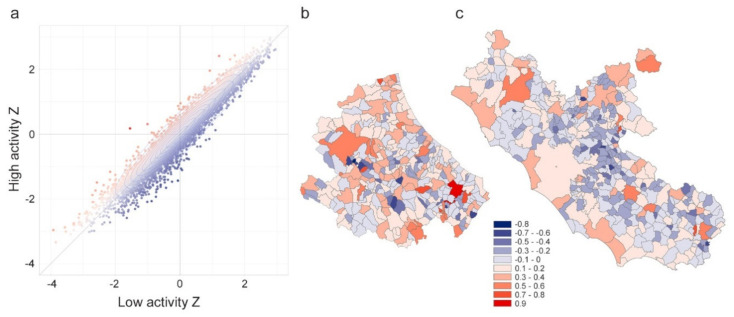
Panel (**a**) include the scatterplot of the Zscores for low-activity time (*x*-axis) and high activity time (*y*-axis). The color of the points codes the difference y–x to diversify the municipalities where incoming commuters prevail over outgoing (red points) from municipalities where outgoing commuters prevail over incoming (blue points). Zscore difference for (**b**) Abruzzi and (**c**) Lazio regions, respectively.

**Figure 7 microorganisms-08-00911-f007:**
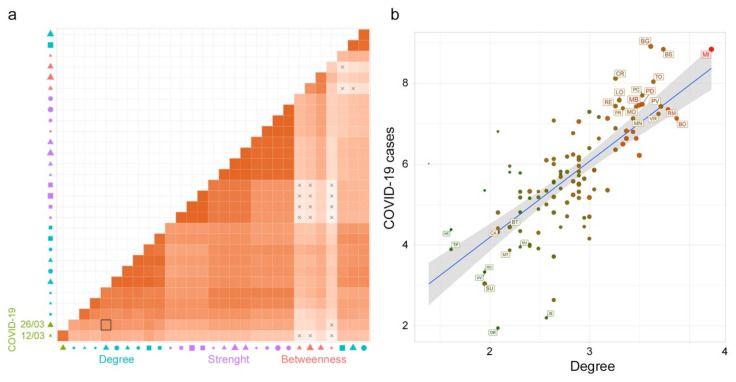
Correlation analysis. (**a**) The heatmap of Pearson’s correlation coefficients among all the variables: Social Network Analysis (SNA) metrics for the rescaled commuting network and subnetworks (with in and out commuters greater than 50, 100 and 1,000), and COVID-19 cases (at different times). The X symbol indicates a nonsignificant correlation. The red scale color indicates a positive Pearson’s coefficient value. In the *x*-axis legend, the symbol color represents the variable groups: COVID-19 cases (green), betweenness (red), degree (light blue) and strength (purple); the symbols: circle, square and triangle represent the directions in terms of in (circle), out (square), and none (triangle); the symbol size from smaller to bigger represents the networks variables (subnetworks with in and out commuters greater than 50, 100 and 1,000 and the whole network). (**b**) A scatter plot graph between Deg50 and COVID-19 cases (as of March 26 2020) in the logarithmic scale. The scale color from green to red is used to characterize the in-strength (from a lower to higher number of incoming commuters) of the node and the symbol size (from smaller to bigger) characterizes the node in terms of out-strength (lower to higher number of outgoing commuters).

**Figure 8 microorganisms-08-00911-f008:**
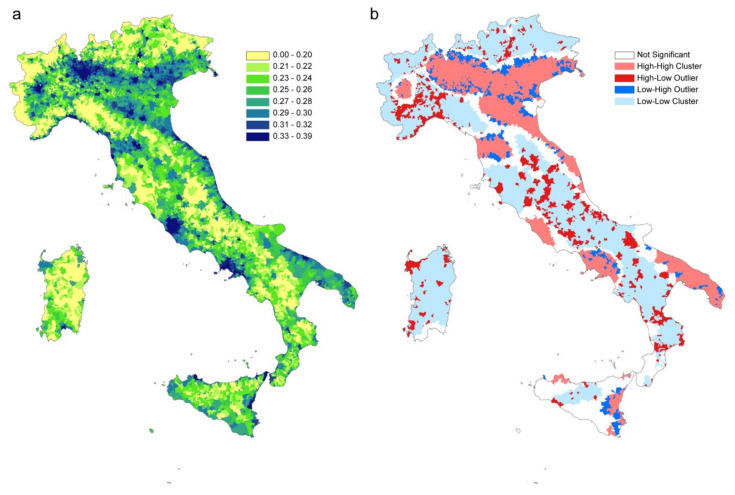
(**a**) β values per municipality (in legend quantile classification). (**b**) statistically significant hot spots in pink (municipalities with high β values in significant clusters), cold spots in light blue (clustered municipalities with low β values); in red municipalities with high β values and surrounded by municipalities with low β values; in blue municipalities with a low β value surrounded by municipalities with high values. Municipalities with no significant clustering or outliers are shown in white.

**Figure 9 microorganisms-08-00911-f009:**
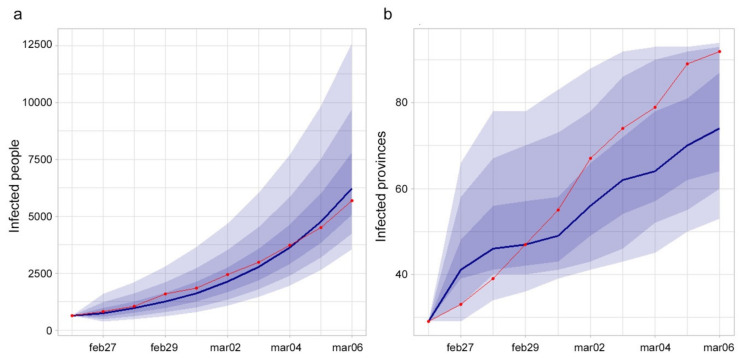
(**a**) Observed (red line) and estimated (blue line and 0.95, 0.80 and 0.50 CI) infected people in Italy. (**b**) Observed (red line) and estimated (blue line and 0.95, 0.80 and 0.50 CI) number of infected provinces.

**Figure 10 microorganisms-08-00911-f010:**
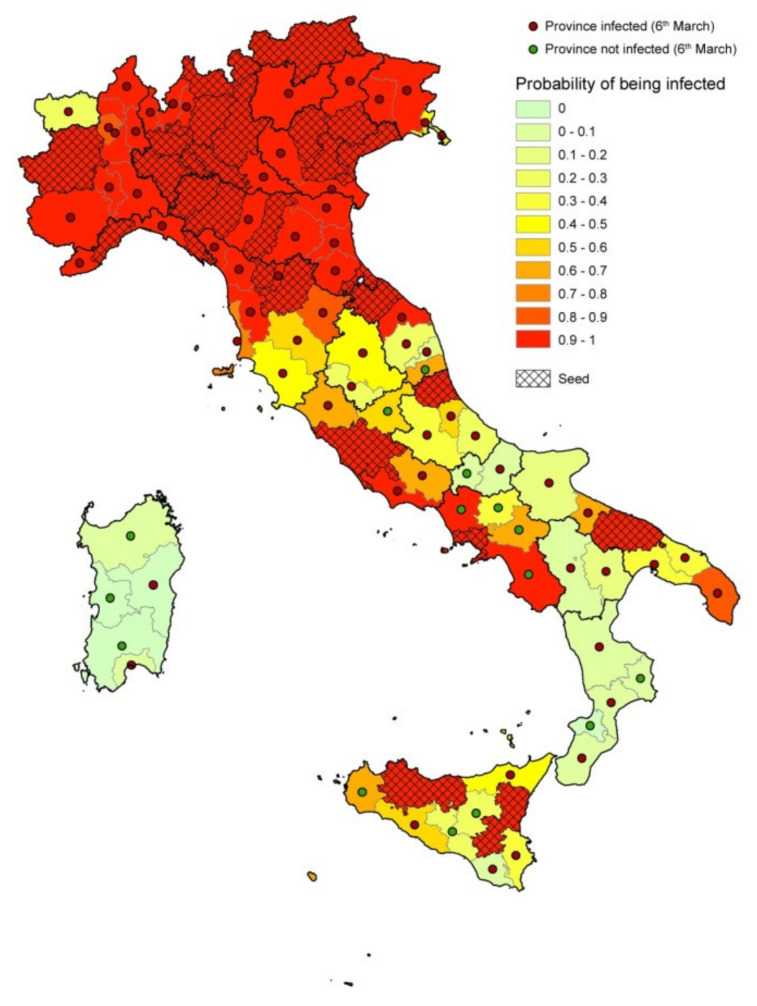
Results of the model. Provinces are colored based on the estimated probability of being infected; red points show the observed infected provinces at the end of the period (February 26–March 6); the green points represent the noninfected provinces at the end of the period; dashed provinces represent the seeds at the beginning of the period (February 26).

**Figure 11 microorganisms-08-00911-f011:**
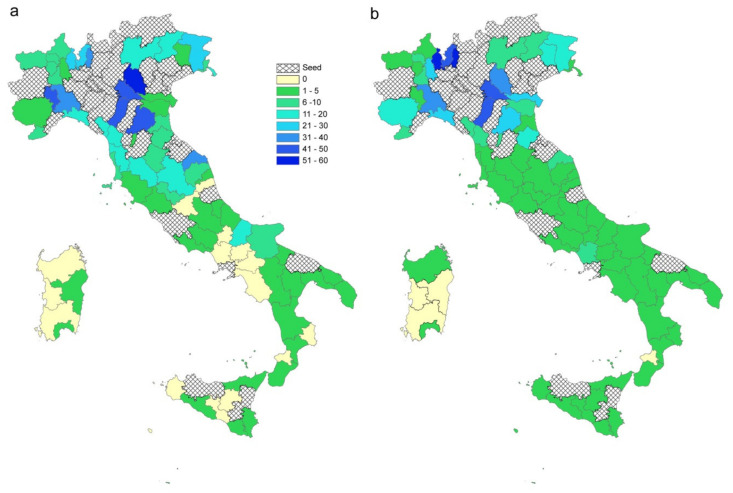
Results of the model. Comparison between observed (**a**) and estimated (**b**) infected people. The upper 95% confidence interval of the estimated infected people is used (when a province is turned into infected during the simulations the value of infected people is considered). Dashed provinces are the initial seeds.

**Figure 12 microorganisms-08-00911-f012:**
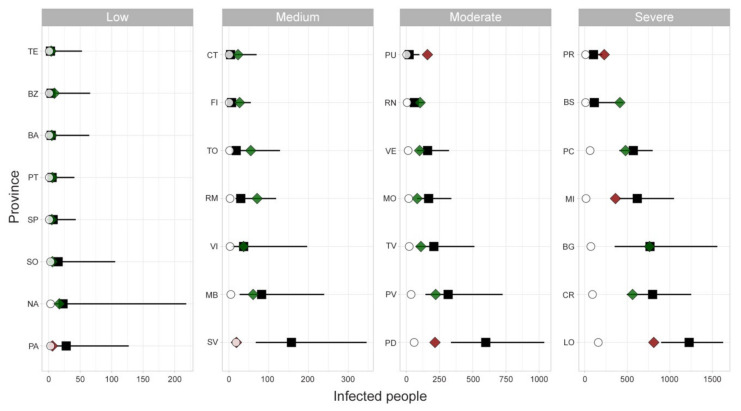
Infected people in seeding provinces as of March 6. Black squares represent median values, lines the 95% confidence interval (CI), green and red diamonds represent observed data within (green) and outside (red) the CI, while white dots represent the observed cases as of February 26. The panels’ order (from low to severe level of infected people) is used for displaying purposes.

**Figure 13 microorganisms-08-00911-f013:**
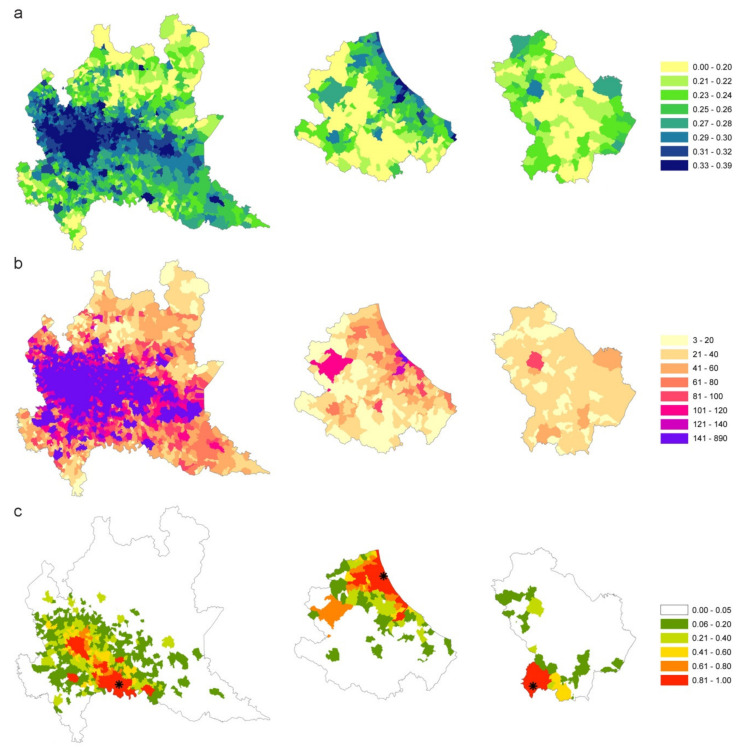
β values (**a**), degree measure (**b**) and estimated probability (**c**) of being infected for each municipality of high (Lombardy), medium (Abruzzi) and Low (Basilicata) affected areas.

**Figure 14 microorganisms-08-00911-f014:**
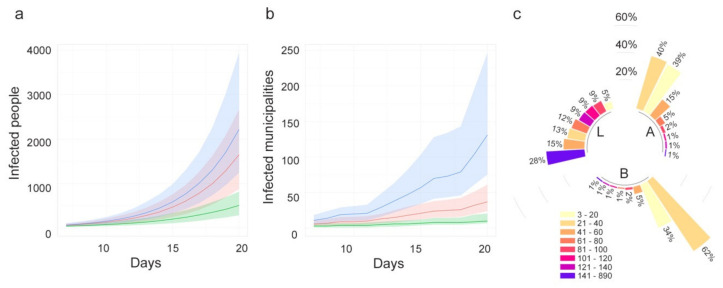
(**a**) Number of estimated infected people (median values and 95%CI) and (**b**) estimated infected municipalities (median values and 95% confidence interval, CI) when the epidemic starts in one seed and lasts 21 days. (**c**). The blue line represents Lombardy, red line is Abruzzi region and green line, Basilicata region. Degree distribution in the three regions: L = Lombardy, A = Abruzzi, B = Basilicata. Bars are in class percentage descending order.

**Figure 15 microorganisms-08-00911-f015:**
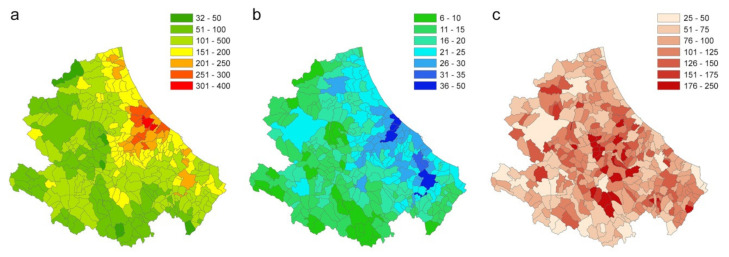
(**a**) Number of infected people (95th percentile) that each municipality causes in the region (excluding itself). (**b**) Number of infected municipalities caused by each seed (95th percentile). (**c**) Ratio between the number of cases caused outside the municipality (95th percentile) and the number of cases caused inside the municipality (95th percentile).

**Table 1 microorganisms-08-00911-t001:** Simulation parameters for each scenario.

Scenario	Num. of Simulation	Num. of Simulation Runs	Deep in Day	Region	Scale Resolution	Seed (t_0_)	I(t_0_)
Scenario 1	1	500	10	Italy	Municipality	Randomly distributed inside the initial infected provinces (29)	Observed cases as of February 26 = 625
Scenario 2	1	500	21	Lombardy	Municipality	Codogno	1
				Abruzzi		Roseto Degli Abruzzi	
				Basilicata		Trecchina	
Scenario 3	305	500	14	Abruzzi	Municipality	All municipalities (305)	1

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
