# Peer review of "A Municipality-Based Approach Using Commuting Census Data to Characterize the Vulnerability to Influenza-Like Epidemic: The COVID-19 Application in Italy"

_microorganisms, 2020, doi:10.3390/microorganisms8060911_

Round 1

Reviewer 1 Report

This is a modeling paper to describe and predict the COVID epidemic in Italy.  Lines 52-58 and 63-71 of the introduction should be in the methods.  Lines 72-74 are elements of discussion.  the model and the analysis are well described. In the discussion, especially model 3, it would be useful to add some modeling about what to expect when travel (and potential transmission events) is increased, and also work through this as inter-provincial and national travel are permitted.  This would test the robustness of the model and be instructive to the readers.

Author Response

Lines 52-58 and 63-71 of the introduction should be in the methods. Done

Lines 72-74 are elements of discussion. Done

The model and the analysis are well described. In the discussion, especially model 3, it would be useful to add some modeling about what to expect when travel (and potential transmission events) is increased, and also work through this as inter-provincial and national travel are permitted.  This would test the robustness of the model and be instructive to the readers.

In scenario 3, although the network is considered closed at region level, the possibility of introducing the infection into any municipality of the region already depends on the assumption that the infection comes from other regions due to potential travelers. At the intra-regional level, traveling is considered free between the provinces, with no restrictions in place. This scenario is then representative of the current situation in Italy, where displacements among regions are now liberalized despite the presence of active cases in many of them.

Scenario 3, therefore, made it possible to map the region's vulnerability to any disease transmission that originates outside the region following the future partial or total easing of the lockdown measures.

We added a sentence in M&M to clarify and stress that Scenario 3 can be used also in case of a new epidemic: “This scenario aims at defining the vulnerability of each municipality to a new epidemic (or a new epidemic wave) and it gives an indication about the potential of each municipality to be the index point of new infections.”

Reviewer 2 Report

In general I liked the paper. However, I think that some aspects are not very clear.

My major concerns:

1) The paper is structured in a confusing way. There is an appendix in which more detailed information on how to tune and provide structure to the different functions appearing in the model is provided. I think that meaningful contributions are given in it, I do not understand very well why to send that information to an appendix (it is not a very technical one).

2) Reading the paper one does not have a clear idea of how the model is fit to data. Some ideas are given, but it is not clear to which extent the paper is just providing indexes that are correlated with risk, or if there is a solid methodology to tune the parameters to obtain a functional model. For example, in lines 157-159 some important key parameters are taken from the literature (references [16] and  [17] ) 

3) I do not think that estimating the probability of detecting cases in a municipality is a realistic setting. The spread of the pandemic is too intense. Maybe it has sense from an index point of view. That is, the author should maybe stress more its functionality as a risk index, more than a "estimation" of the probability of detecting cases.  

Minor:

a) Social Network Analysis (SNA) centrality measures are used (line 52 and appendix). I would provide a reference or provide more detail. 

b) In line 80 the author claim that they have adjusted the geographical data set after 2011. I would detail more if this just consisted of updating the population, or something more involved was required.

c) In relation to open data sources in the context of Covid-19 I would cite

Teodoro Alamo, Daniel G. Reina, Martina Mammarella and Alberto Abella. Covid-19 Open-Data Resources for Monitoring, Modeling and Forecasting the Epidemic. Electronics, 2020.

d) Reference 7 is badly written. 

e) In relationship with Italy, I think that it would be interested to cite

Giordano, G., Blanchini, F., Bruno, R. et al. Modelling the COVID-19 epidemic and implementation of population-wide interventions in Italy. Nature Medicine (2020).

f) In 119 there is an expression for the population that is not consistent with the one in the appendix (see line 517).  

g) In line 215 it is said that the number of plausible cases is 10 times larger than the one reported by the administration. I guess that this factor can now be better estimated with the already serology studies published for Italy.

In general, I liked the paper.  However, I believe that the authors could improve the readability of the paper.

Author Response

My major concerns:

  1. The paper is structured in a confusing way. There is an appendix in which more detailed information on how to tune and provide structure to the different functions appearing in the model is provided. I think that meaningful contributions are given in it, I do not understand very well why to send that information to an appendix (it is not a very technical one).  We thank the reviewer for the valuable suggestion. The appendix text is now integrated into the paper, in lines154-242.                                       
  2. Reading the paper one does not have a clear idea of how the model is fit to data. Some ideas are given, but it is not clear to which extent the paper is just providing indexes that are correlated with risk, or if there is a solid methodology to tune the parameters to obtain a functional model. For example, in lines 157-159 some important key parameters are taken from the literature (references [16] and  [17])                                             We understand that the text might induce to confusion and we tried to clarify. We only consider two epidemiological parameters, the number of (undetected) cases per each officially detected case (K) and the infection contact rate. K is taken from the literature and the infection contact rate is estimated. We clarified that we started from the estimation of the effective infection contact rate β0 using the epidemic data and then we fine-tuned it including information on population and activity time (to obtain a functional model). The fine-tuned β has then be mapped to show differences among municipalities to capture disease permeability and at the same time it is used in the SI model to estimate space-time spread of the epidemic.
  3. I do not think that estimating the probability of detecting cases in a municipality is a realistic setting. The spread of the pandemic is too intense. Maybe it has sense from an index point of view. That is, the author should maybe stress more its functionality as a risk index, more than a "estimation" of the probability of detecting cases.                       We agree with the referee that  any modeled epidemic is a simplification of what happens in reality. Indeed, we consider the estimation of cases that can arise from a municipality due to local mobility as an expression of the risk of this municipality to be the start of new infections. At the end of the discussion we reported (lines 549-552): "This approach is particularly useful not only in the beginning of the epidemic but also in the last phase, when the risks deriving from the gradual lockdown exit strategies must be carefully evaluated. In fact, the major risk in this latter phase is the resurgence of infection transmission through a progressive re-opening of the productive systems". However, we better explain this aspect in M&M, when we describe the scenario 3: "Local spread in the Abruzzi region is simulated (during the first 14 days of the epidemic) considering each municipality as a seed for each simulation. This scenario aims at defining the vulnerability of each municipality to a new epidemic (or a new epidemic wave) and it gives an indication about the potential of each municipality to be the index point of new infections. The choice of the Abruzzi region has only an illustrative purpose. We assume that the infection starts in turn in each municipality so to assess the weak points of the whole region”.

 Minor:

  • Social Network Analysis (SNA) centrality measures are used (line 52 and appendix). I would provide a reference or provide more detail. 

The SNA is used only as a preliminary study to demonstrate the correlation between commuting data (expressed through centrality measures of the network) and COVID-19 cases. Actually references 15, 16, and 17 were already present in the paper to explain the centrality measures adopted and their meaning in disease transmission studies. However we added reference 18 to complete the list of all the centrality measures calculated in the study.

In addition we clarified the use of SNA in the text in the lines 58-60: “The study is organized to first evaluate the use of commuting data as risk factors to COVID-19 spreading in Italy, calculating Social Network Analysis (SNA) centrality measures at province level and performing a data correlation analysis”.

  • In line 80 the author claim that they have adjusted the geographical data set after 2011. I would detail more if this just consisted of updating the population, or something more involved was required.

Being a simple adjustment of population and geographical boundaries, authors believe it is not strictly necessary to further explain. To clarify to the referee: we used the data collected in the 2011 census data in Italy. This commuting dataset contains 8.094 municipalities. The geographical file currently in use for the Italian territory in 2020 contains 7,915 municipalities (several municipalities disappeared and were merged to others). In order to have a correspondence between the geographical file and the 2011 census data file we spatially aggregated data on the commuting population. The final dataset contains 7,915 municipalities with a population of residents of 60,340,328 and 28,805,440 as reported in the commuting dataset. Therefore, it just consists of updating the population.

  • In relation to open data sources in the context of Covid-19 I would cite Teodoro Alamo, Daniel G. Reina, Martina Mammarella and Alberto Abella. Covid-19 Open-Data Resources for Monitoring, Modeling and Forecasting the Epidemic. Electronics, 2020. Done
  • Reference 7 is badly written. Corrected
  • In relationship with Italy, I think that it would be interested to cite Giordano, G., Blanchini, F., Bruno, R. et al. Modelling the COVID-19 epidemic and implementation of population-wide interventions in Italy. Nature Medicine (2020). Done
  • In 119 there is an expression for the population that is not consistent with the one in the appendix (see line 517). Authors don’t see the inconsistency reported by the referee. Probably the use of “resident population” is misleading, but it is necessary to identify people living in a place during the low-activity period. Or probably the referee sees as misleading to name the municipality “C”  while in line 517 we refer to municipality “L”. The two figures represent different aspects that’s why we used different letters. If this was not the referee’s concern we are of course available to further suggestion/correction
  • In line 215 it is said that the number of plausible cases is 10 times larger than the one reported by the administration. I guess that this factor can now be better estimated with the already serology studies published for Italy. Although recent studies may give a better estimate of this factor thanks to serological tests data availability (that in any case are extremely variable in results and region of testing), we should rerun all of the simulation of each scenario to account for an updated value that is not yet an accurate estimate of the Italian situation. In addition, considering the the proportionality, correlation and patterns (and then our findings and conclusion) will be the same even using different values. For this reason, authors added the following in D&C: ”Indeed, we only consider two epidemiological parameters, the number of (undetected) cases per each officially detected case (K) and the infection contact rate β. We made the assumption of K being equal to ten [20]. Although more accurate estimation could soon be available, our results in terms of correlation and patterns are not affected by this parameter because it is a proportionality factor.”

Round 2

Reviewer 1 Report

Accept as revised